# Lung Volume Calculation in Preclinical MicroCT: A Fast Geometrical Approach

**DOI:** 10.3390/jimaging8080204

**Published:** 2022-07-22

**Authors:** Juan Antonio Camara, Anna Pujol, Juan Jose Jimenez, Jaime Donate, Marina Ferrer, Greetje Vande Velde

**Affiliations:** 1Preclinical Therapeutics Core, University of California San Francisco, San Francisco, CA 94158, USA; 2Onna Therapeutics, 08028 Barcelona, Spain; pujolesclusa@gmail.com; 3Preclinical Imaging Platform, Vall d’Hebron Institute of Research, 08035 Barcelona, Spain; juanjosejimenez@vhio.net (J.J.J.); jaimedonate@vhio.net (J.D.); 4Gnotobiotics Core Facility, University of California San Francisco, San Francisco, CA 94158, USA; marina.ferrerclotas@ucsf.edu; 5Biomedical MRI/MoSAIC, Department of Imaging and Pathology, Faculty of Medicine, KU Leuven, 3001 Leuven, Belgium; greetje.vandevelde@kuleuven.be

**Keywords:** microCT, preclinical imaging, lung, volume, quantification

## Abstract

In this study, we present a time-efficient protocol for thoracic volume calculation as a proxy for total lung volume. We hypothesize that lung volume can be calculated indirectly from this thoracic volume. We compared the measured thoracic volume with manually segmented and automatically thresholded lung volumes, with manual segmentation as the gold standard. A linear regression formula was obtained and used for calculating the theoretical lung volume. This volume was compared with the gold standard volumes. In healthy animals, thoracic volume was 887.45 mm^3^, manually delineated lung volume 554.33 mm^3^ and thresholded aerated lung volume 495.38 mm^3^ on average. Theoretical lung volume was 554.30 mm^3^. Finally, the protocol was applied to three animal models of lung pathology (lung metastasis and transgenic primary lung tumor and fungal infection). In confirmed pathologic animals, thoracic volumes were: 893.20 mm^3^, 860.12 and 1027.28 mm^3^. Manually delineated volumes were 640.58, 503.91 and 882.42 mm^3^, respectively. Thresholded lung volumes were 315.92 mm^3^, 408.72 and 236 mm^3^, respectively. Theoretical lung volume resulted in 635.28, 524.30 and 863.10.42 mm^3^. No significant differences were observed between volumes. This confirmed the potential use of this protocol for lung volume calculation in pathologic models.

## 1. Introduction

Since their discovery, X-rays have become a source of knowledge and a base for development of imaging technologies such as computed tomography (CT), which meant a significant step forward in medical imaging by delivering three-dimensional images that facilitated diagnosis of different pathologies. CT is currently considered as a gold standard in clinical lung and thorax examination and is increasing its relevancy in preclinical research because of its capability to obtain information in a non-destructive way, compared to histopathology and other post mortem analyses. Parallel with the increased use of X-rays in preclinical research, the imaging technology has been adapted to the smaller size of the subjects with a focus on increasing the resolution within reasonable limits of delivered X-ray dose. While large animals such as sheep and pigs are frequently scanned in clinical devices, mice, rats or even rabbits are examined in dedicated systems, adapted to their anatomy and physiology. Preclinical microCT scanners provide higher spatial and temporal resolution than their clinical counterparts to be able to deliver information on a scale relevant to small animals. Moreover, recent technological progress has reduced the levels of radiation applied to the animals during microCT acquisitions and consequently, most of the collateral effects of radiation in biologic structures can be prevented [1,2], leading to the possibility of longitudinal studies in the same animal for understanding of how pathologies evolve.

While the use of imaging technologies in clinical medicine is principally based on qualitative evaluation of images, imaging in basic and preclinical research cannot be separated from data quantification and statistics in order to obtain scientific conclusions. This fact leads to a significant increase in the amount of data and time required for the process of image acquisition, processing and analysis. Regarding microCT studies, information can be obtained from the images based on the voxel characteristics that can be analyzed: radiodensity (expressed in grey scale or Hounsfield units) and three-dimensional location. The selection of a specific group of voxels in the image (called a region/volume of interest or ROI/VOI) is based on one or both of these characteristics: While attending to the voxel density, we could apply thresholding and plot histograms; while regarding spatial location, we could refer to frontiers, limits and edges. In most of these approaches, quantification is time-consuming and the need for highly trained human resources and time generates a bottleneck in the projects that include CT analysis.

Imaging of the respiratory system is one of the main applications of X-rays in clinical and preclinical imaging. Planar X-ray images and CT scans are frequently used for lung exams due to their capability to discern between radiolucent and radiodense structures. Lung tissue is basically composed of air and low dense soft tissues and they are poor X-ray absorbents (radiolucent), so graphically they appear as grey-to-black regions. Meanwhile, lung edema, ribs, heart and muscles are dense structures which absorb X-rays in a higher quantity, resulting in bright grey-to-white structures (radiodense). Lung pathologies could affect the radiodensity of lungs in both ways: reducing the density, such as in emphysema and hyperinflation, or increasing it, such as happens in case of lung tumors, pneumonia or inflammatory and fibrotic processes. In these cases, the definition of the limits between radiodense lung tissue and adjacent structures such as muscles or heart could be challenging.

Aerated lung volume is a parameter frequently quantified to analyze the development of respiratory illness. Different approaches have been designed for measuring aerated lung volume in microCT images. They are principally manual and thresholded segmentations, followed by others such as region growth-based analysis, all based on the hyperlucent nature of aerated lung voxels.

Manual segmentation of the lungs could be described as drawing lung limits in different microCT slices and composing a 3D VOI [3,4,5,6,7,8,9]. This process is time consuming and operator dependent [5,9,10,11], and is challenging during total lung volume calculation in pathologic animals, where lung tissue becomes as radiodense as surrounding soft tissue. Several studies have measured the potential error of this technique with varying results [5,9,12]. Furthermore, manual delineation considers intrapulmonary vessels and other no-pulmonary structures (e.g., lymphatic structures and big airways) as lung tissues due to the complexity of avoiding intrapulmonary structure selection during manual delineation [4,7,11,13].

A second method for aerated lung segmentation is the isocontour threshold, an automatic computed process that selects the voxels included in a previously defined threshold or range of intensities (intensity window). This method is semiautomatic, faster and theoretically non-operator dependent [14,15,16,17,18,19,20,21]. On the other hand, this approach depends on the radiodensity of the structures, which can change frequently in lung pathologies [8]. Increasing the intensity window for including the pathologic lung tissue could lead to an inclusion of non-pulmonary soft tissue and an overestimation of total lung volume. For this reason, manual delineation is usually used as the gold standard for total lung volume calculation [6,21,22,23,24] as automatic methods derail in the presence of radiodense lung pathology.

A third approach is the region-based method, which manually defines a specific region for automatic growing of a VOI, limited by a threshold [10,20,24,25]. This method is an evolution and combination of the previous ones with a higher accuracy but similar disadvantages.

Due to the high need for objective and high-throughput quantification methods, other methods are constantly under development to address lung segmentation, including automatized and machine learning protocols [16,17,21,26] or combined procedures [25,27,28]. These methods are automatizations of the first ones, technically restricted and not always available. Furthermore, they need large annotated input datasets for prior network training that requires segmentation of the lung by one of the available methods as outlined above.

To address the need for a simpler and faster method of lung volume calculation for preclinical research, we defined in this work a mathematical approach that is based on thorax anatomy and distance measurements in microCT studies. The protocol is validated in a group of naïve animals and later applied to three different lung pathology models. This method would alleviate an unmet need during lung evaluation of microCT images as the quantification of lung volume is completely automatic, with the consequent saving of time and reduction of variability.

## 2. Materials and Methods

### 2.1. Animal Models

Four different mouse models were included in this study: healthy animals (*n* = 37), metastatic lung tumor model [29] (*n* = 23), transgenic primary lung tumor model [30] (*n* = 18) and a fungal lung infection model [1] (*n* = 9). Mouse strains were C57/BL-6 (healthy, metastatic lung tumor and transgenic lung tumor groups) and BALB-C (fungal infection). The healthy model was defined as control group for validation of the protocol. For each pathologic model, sham-induced age- and gender matched control animals were included. The distribution of the animals was as follows: Metastasic lung tumor model (12 pathologic, 11 negative control), transgenic primary lung tumor model (8 pathologic and 10 negative controls) and fungal lung infection model (7 pathologic and 2 negative controls).

### 2.2. Image Acquisition, Data Quantification and Statistical Analysis

Two dedicated preclinical microCT scanners were used for image acquisition: Quantum Fx (Perkin Elmer, Waltham, MA, USA) and Skyscan1076 (Bruker, Kartuizersweg 3B. 2550 Kontich, Belgium). Quantum Fx was used for healthy and oncologic models (metastasic lung tumor and transgenic lung tumor model) and the data of the fungal infection model was acquired on the Skyscan1076. All animal experiments were performed according to European and local legislation and approved by the appropriate local ethics committee. Acquisition parameters were, for Quantum Fx: 90 Kv/160 µA X-ray energy, 2 min of static scanning with a rotation step of 0.1 degrees and 30 frames per second for 3672 final projections. Reconstructed studies were 512 × 512 × 512 pixels of 0.050 mm size in 16-bit images. Skyscan acquisition parameters were defined as 50 kV/200 µA of X-ray energy, resulting in 14 min scanning with a 0.7 degrees rotation step in reconstructed 16-bit images of 1024 × 1024 × 1024 pixels and 0.035 mm pixel size.

Mice were anaesthetized with inhaled isoflurane (Aerrane. Baxter Lab. Deerfield, IL, USA), 3% in fresh air during induction and 1.5% in maintenance and placed in a standardized supine position on the scanning bed. Free respiratory movements were allowed during the acquisition process. The studies from the Quantum Fx (healthy, metastatic and transgenic models) were acquired in static mode. The fungal infection model was acquired in the Skyscan system and the analyzed images correspond to end-expiration phase.

Both microCT systems were calibrated for Hounsfield Units (HU) using phantoms. In the Quantum Fx system, the phantom consisted out of a 5 mL plastic tube filled with 2 mL of water and 3 mL of air. For the SkyScan1076 the phantom consisted in 1.5 mL air in a 50 mL tube filled with water as described before [31]. The calibration protocol was carried out by creating two VOIs placed in water and air. The average grey scale value for air VOI was associated to −1000 HU while the grey value for water was assigned to 0 HU.

Microview (Microview© Parallax Innovations, Ontario, Canada) software was used for thoracic volume calculation and manual delineation of the lung. Amide (Amide Software© Andreas Loening, Los Angeles, CA, USA) was run for thresholded volume. Upon prior quality control, scans with artefacts (due to movement, mainly) were repeated upon acquisition. These artefacts consisted mainly in movements of artifacts that created intrapulmonary shadows from the bone structures and could potentially affect the intensity values of the pixels. No post-processing modifications were applied on the images (e.g., filtering or smoothing) in order to maintain the original voxel intensity and heterogeneity values.

Sigmaplot (Systat Software Inc. San Jose, CA, USA) software was used for statistical analysis. All the results passed a normality test (Shapiro-Wilk) for normal distribution confirmation before subsequent work. The correlation analysis was performed based on the Pearson Product Moment Correlation protocol, while the comparisons between groups and values were performed using t-test analysis with a 95% confidence interval for difference of means.

### 2.3. Thoracic Volume Calculation

The protocol started with a reorientation of the microCT image. Vertebrae and sternum were aligned in the axial plane as well as contralateral ribs in coronal view. This reorientation of the data was essential to obtain reproducible results.

The thorax volume was calculated based on its similarity with a truncated cone with a minor base at the thoracic carina level and big base at the diaphragmatic cupola level (Figure 1). The formula for truncated cone volume is: **Volume = PI/****3**
**×**
**eight × R1^2^ × R2^2^ × R1 × R2**, where R1 is the small base of the cone, and R2 is the large base and height corresponds to the distance between the bases. The small base of the cone is measured at the carina level (principal bronchus division) in coronal view, from the external lung limit to the contralateral one (Figure 1). In case no lung area was observable at this level, the distance between the internal surfaces of the ribs was used. The large base was measured at the diaphragmatic level in sagittal view and was defined as the distance between the ventral aspect of the thoracic vertebrae and the abdominal wall. The height of the cone was measured either in sagittal or coronal views (Figure 1).

### 2.4. Manual Delineation and Thresholded Total Lung Volume

Manual delineation of total lung volume was performed as described in [32]. In brief, manual drawing was carried out in several slices, starting at the carina level, delimiting the frontier between lung tissue and other structures such as heart or ribs (Figure 2). It finished with the last lung tissue slices. This procedure was followed by an automatic interpolation between slices. A later manual correction was carried out if necessary.

An automatic threshold was applied to segment aerated lung volume. This thresholding was performed using two different intensity windows, based on previously published values [8,12,20,23] and included the ranges (−700 to −300 HU) and (−700 to −400 HU). After verifying the accuracy of the segmentations for mouse lungs (Figure 2), the (−700 to −300 HU) intensity range window was chosen as a reference.

### 2.5. Theoretical Lung Volume Calculation

A lineal correlation between thoracic volume and manually delineated total lung volume was obtained later. This correlation generates a mathematical formula that connects both measurements. Applying retrospectively this formula to the thoracic volume, a theoretical lung volume was obtained. This theoretical lung volume (goal of the whole protocol) was then compared to the manually delineated lung volume in order to evaluate their similarities.

### 2.6. Statistical Comparison between the Different Volumes

Volume ratios were calculated to compensate for the potential differences in animal model or size that could affect the raw values of thorax and lung volumes. These assumptions are based on the rationale that healthy animals should have similar ratios in aerated lungs and intrathoracic soft tissue structures independently of the animal model or size. These ratios were: Manually delineated volume/thoracic volume; thresholded volume/manually delineated volume and thresholded volume/thoracic volume.

Different statistical analyses were run for evaluating the changes in lung volumes and the feasibility of the designed protocol in different scenarios. The lung volumes were compared between sham-control negative animals and the pathologic one in each animal model. Thoracic and manually delineated volumes were compared for evaluating a possible variation of total lung volume due to the presence of lung pathology. Thresholded lung volume was compared to ensure the presence of the pathology. Finally, the theoretical lung volume was compared to the gold standard to confirm the protocol’s independence from the existence of a lung pathology. For this, the statistical deviation of the theoretical lung volume from the gold standard was calculated.

### 2.7. Reproducibility and Repeatability (R&R) Analysis

R&R analysis was carried out to evaluate the reliability of the protocol. The repeatability analysis was performed by two operators (A.P., J.J.J.) that measured the same images 3 times in a blinded way. For reproducibility analysis, the volumes were calculated in the same batch of data by the same operators (A.P., J.J.J.) and results were compared between operators.

### 2.8. Specificity and Sensitivity Analysis

With the aim to clarify the potential use of the protocol, a specificity and sensitivity analysis was applied to the results. For each pathologic group, a population lung volume was calculated as the mean value of the sham-negative control animals. The different studies were compared to this value and the percentage of deviation from this population value was obtained. In the pathologic studies, a percentage of deviation over 5% classified the study as true positive, while a value below 5% classified the study as false negative. In the same way, for the sham-negative control animals, a percentage over 5% was regarded as a false positive and a value below 5% was defined as a true negative.

### 2.9. Validation of the Protocol

Prior to its application in pathologic models, a validation analysis was performed in a group of controls, i.e., healthy animals. For this purpose, thoracic lung volume, manually delineated lung volume and thresholded segmented aerated lung volume were calculated in each study. After that, the regression formula that connects thoracic and manually delineated volumes was obtained. With this formula, a theoretical lung volume was calculated and ratios between volumes were obtained too.

The protocol would be validated only if the theoretical lung volumes obtained from the regression formula were statistically similar to the gold standard values (manually delineated lung volumes). With this aim, a Bland-Altman analysis was applied to the results, assessing the differences between these two volumes (theoretical and manually delineated). This analysis consists of the statistical evaluation of the differences between the theoretical lung volumes obtained and the gold standard volumes (manually delineated lung volumes). After the subtraction of these volumes, they are plotted with the mean value of this difference and the range of values for a 95% of confidence.

## 3. Results

### 3.1. Feasibility of the Protocol for High-Throughput Applications

Lung microCT data acquisition happened with a very minor number of incidents, resulting in only 4–5% of scans that needed to be repeated instantly due to movement artefacts in the images. Image acquisition is feasible within approximately 2 to 14 min per animal, depending on the imaging equipment.

To evaluate the effectiveness of the subsequent lung analysis protocol, we measured the time a trained operator needed to perform each analysis per scan. The examiner loaded the file, rotated the images until the proper position and made the different measurements for thoracic volume calculation. This process took 2 to 3 min per scan. Manual lung delineation took a trained operator fifteen to twenty minutes per scan, depending on the extent of lung pathology. Threshold analysis cost five to ten minutes per study, depending on the affectation of the lung tissue. The linear regression formula for each animal model takes 2–3 min and once it is calculated, the theoretical lung volume is obtained in 4 min (3 for thoracic volume calculation plus 1 more for applying the regression formula).

We can reduce the time consumed for obtaining the lung volume from 15–20 min with manually drawing to 4–5 using this new protocol.

### 3.2. Reproducibility and Repeatability Analysis of the Protocol

Reproducibility and repeatability analyses were performed comparing results between two experienced operators (reproducibility) and between consecutive analyses in the same study by one single experienced operator (repeatability). Results are displayed in A2. No significant differences were observed between operators (*p* = 0.791) after a Tukey test (All Pairwise Multiple Comparison). According to the repeatability analysis based on a Gage R&R ANOVA test, the contribution of operator variability to total variation reached 2.98% in the thoracic volume calculation, 1.04% in manual delineation of lungs and 0% in the automatic threshold volume calculation. Attending to these statistical values, we consider all protocols as repeatable and reproducible.

### 3.3. Specificity and Sensitivity Analysis of the Protocol

The general specificity of the protocol was 78.26% and the sensitivity was 74.07%. Looking at the different models, for the metastatic model the sensitivity was 66.67% and specificity 81.82%. In the transgenic model, they were 74.07 and 78.26%, respectively. Finally, the fungal infection model was 100% in both sensitivity and specificity.

### 3.4. Validation Analysis of the Protocol in a Group of Healthy Mice

Results are displayed in Table A3 and Figure 3. The mean value of thoracic volume was 887.45 mm^3^ (standard deviation, 192.47 mm³) while manually delineated lung volume was 566.67 mm^3^ (S.D., 88.40 mm^3^) and thresholded volume was 495.38 mm^3^ (S.D., 88.92 mm^3^). There are significant positive correlations between the three volumes (*p* < 0.05). The ratio “manually delineated volume/thoracic volume” was 64.83% (S.D. 6.87 mm^3^) while “threshold volume/manually delineated volume” was 87.25% (S.D. 3.29 mm^3^). Finally, the “threshold volume/thoracic volume” ratio was 56.45% (S.D. 5.27 mm^3^). The R^2^ values of the different correlations were 0.95 for “threshold volume/manually delineated volume” and 0.68 for “threshold volume/thoracic volume”.

The regression formula was obtained later, and its mathematical expression was: Y = 0.1955X + 298.07, where X was the measured thoracic volume. The R^2^ value was 0.7126. Applying this formula, the mean value for theoretical lung volume was 554.30 mm^3^ (S.D. 81.84 mm^3^). Comparing this theoretical lung volume to manually delineated volume, there is a significant correlation between the values (*p* < 0.05), with a value of 0.85 for R^2^ in the correlation analysis.

Because the correlation between theoretical and manually delineated lung volumes is statistically significant (*p* < 0.05), we can conclude that calculating indirectly the lung volume from thoracic volume in microCT studies is a valid protocol that returns trustable values in a short time. This conclusion is reinforced by the results from the Bland-Altman analysis (Figure 3).

### 3.5. Applicability of the Protocol in Different Mouse Models of Lung Pathology and Different Scans

We next set out to apply the validated protocol in animal models of different lung pathologies to evaluate the applicability of the process in pathologic animals. For this purpose, we reproduced the protocol in three animal models of lung pathology: metastatic lung tumor, transgenic lung tumor and fungal infection (Figure 4). In each model, we defined a group of sham-negative control animals as the reference. These controls would help to compensate for the potential intergroup differences when comparing results between animal models.

The fungal infection model was acquired on a different microCT device, as described before. There were no differences in the different statistical results between microCT systems.

Results of pathologic models are displayed in Table 1 and Table A3 and Figure 5.

The correlation between thoracic volume and manually delineated lung volume in these sham-negative control animals was statistically significant (*p* < 0.05). For the theoretical lung volume, comparing with the measured manually delineated lung volume, no statistical differences were obtained in any of the groups (*p* < 0.05). The results of the negative control groups followed the trend observed in the validation group.

In pathological animals, comparing the theoretical lung to the manually delineated lung volume, no statistical differences were obtained in any of the groups. The theoretical lung volumes were statistically similar to the manually delineated volumes, used as gold standard values.

Comparing the different volume results between sham-negative control and pathologic animals in each model, the only significant differences were between volumes observed in the metastatic lung tumor model (threshold lung volume). In the rest of comparisons, the thoracic, manually delineated and theoretical volumes in pathologic animals are higher than the sham-negative control animals. On the other hand, the threshold volume is reduced in pathologic animals compared to negative control values. All these values are displayed in Table A3.

A percentage deviation from the gold standard was calculated to compare the accuracy of the theoretical lung volume with the gold standard in the pathologic groups. The results for the metastatic lung cancer model were the following: 6.72% deviation in the pathologic group and 3.68% in the sham-negative control group; in the transgenic model, 7.06% in the pathologic group and 7.32% for the negative control, and in the fungal infection model, 8.99 and 17.11%, respectively.

Grouped as a single experiment, the mean deviation of the values between theoretical lung volume and manual delineation in pathologic animals reaches 7.41%, while the negative control animals from the same animal models have a mean deviation of 6.43%, bringing to light the independence of theoretical lung volume from the presence or absence of lung pathology. This is another point that supports the applicability of the presented protocol in lung pathology research. At this point, we can confirm that the protocol and the theoretical lung volume calculation can also be applied in pathological animals.

Comparing between equipment, no significant differences were observed between the fungal infection model and the rest of the models. The volumes and ratios in the sham-negative control animals are not significantly different. For comparing pieces of equipment, only the negative animals were compared to avoid a potential bias of the pathology in the analysis.

## 4. Discussion

MicroCT data quantification is a key point in lung pathology research, especially for lung volume calculation. This parameter is extensively used for lung capacity evaluation, as well as for drug efficacy assessment in different pathologies such as lung fibrosis and similar. Different attempts to obtain lung volumes indirectly from anatomical references have been tried previously in human medicine with irregular results. In the mid-80s, Cooper and collaborators measured lung volumes in human patients using a mathematical approach from X-ray films. They developed a protocol that combined manual drawing of lung limits on 1 cm thick slices of X-ray films with mathematical integration of the complete batch of slices that composed the lungs. It was a primitive approach to manual lung delineation that we can make nowadays with CT or microCT and dedicated software [4], and the first attempt at addressing the lung volume calculation using a geometric approach. In our manuscript, a novel and fast method for thoracic volume calculation is examined using three measurements defined by anatomical landmarks. The theoretical lung volume is obtained later, applying a previously calculated regression formula.

Friedman et al. measured lung volumes with CT images and, as a novel idea, made a compensation for the results using patient weight [9]. This was, as far as we know, the first attempt to introduce a correction or compensation for patient size in the results. Following this idea, in our protocol, we calculate different ratios between volumes in order to neutralize the effect of animal size in the results. We can describe the complete thoracic volume as a combination of the manually delineated lung volume plus other intrathoracic volumes such as heart, big vessels, intercostal muscles or ribs. In healthy animals, the proportion of lung volume in the complete thoracic volume is maintained, as can be assessed by the “manually delineated lung volume/thoracic volume” ratio. In a similar way, manually delineated lung volume is composed of threshold lung volume plus intrapulmonary vessels and other structures whose densities are out of the selected radiodensity range. The fact of constant proportions of structures in the thorax is supported by the positive correlations between volumes in our results and has been previously described in the literature [14]. In their publication, Barck et al. manually segmented the heart and lung volumes in a batch of studies and created an automatized tool for lung and heart volume calculation. During the validation phase, they observed steady and delimited values for both structures. In our study, the ratios between the different volumes were stable and consistent for all the negative control groups and the healthy animals. This means that, as published by Barck, the intrathoracic structures keep a permanent correlation of partial volumes in the thorax cavity.

In order to evaluate the accuracy of the mathematical process for lung volume calculation, we made a comparison between the theoretical lung volume, obtained indirectly from the thoracic volume using a regression formula, and the manually delineated lung volume, obtained from the image analysis. This estimated the precision of the mathematical approach. The agreement between the two methods was analyzed in two ways. Both linear regression and Bland-Altman analysis allowed us to conclude that the lung volume can be calculated indirectly based on the thoracic volume.

For assessing the accuracy of our protocol, we decided to use manual segmentation as the gold standard, but also compared our results with threshold segmentation values. For threshold volume calculation, the reference HU window (−700/−300) was based on visual evaluation and has been used in previously published research [8,12,20,23], according with the standard values for lung tissue HU (mean density −500 HU) [6,17,18,27]. Different publications used the same gold standard due to its stability (automatic segmentation) and fast procedure [12,25]. On the other hand, this density window only includes mixed air/tissue voxels in the segmentation, both pure air and soft tissue voxels (e.g., pathologic lung tissue) will be excluded from the selection. Therefore, threshold volume calculation should be discarded in pathologic models when trying to measure complete lung volumes because lung density changes will follow errors during automatic segmentation. In these situations, manual delineation should be considered as the reference value, increasing significantly the time required for quantification and adding operator dependency to the total bias of the experiment. Attending to this lack of an easy, fast and potentially applicable method in pathologic studies, we designed the protocol presented and evaluated in this work. After comparing the novel protocol results with the gold standard (manually delineated segmentation), there were no significant differences between protocols attending to the results obtained. The presented novel analysis takes 4–5 min, significantly shorter than the manual drawing, which frequently takes 15 to 20 min.

Regarding the reproducibility and repeatability of the protocol, the contribution of operator variability to total variation reached 2.98% in the thoracic volume calculation. No significant differences were observed between operators in the volume calculations. According to our results, we can consider that the calculation of thoracic volume is consistent. Comparing other authors, Joskowicz et al. found no operator dependency during manual delineation of different structures in clinical CT, but pulmonary volume variability reached 10% between examiners [5]. Shin et al. made an analysis of repeatability in automatized threshold-based lung volume calculations and no differences were found between examiners, but a lower repeatability in lung volumes was observed in pathologic groups [12]. Xin and collaborators achieved 96% repeatability and 93% reproducibility with a semiautomatic segmentation of injured rat lungs [8]. Finally, Zhu et al. did not find inter-observer variability during manual lung delineation [9] but variability appeared in other thoracic structures such as the heart and esophagus, probably because of the poor soft tissue contrast between anatomical structures. We consider that our analysis is in concordance with the previously published findings regarding operator variability.

Manual lung delineation is defined as an operator-dependent technique and, in order to reduce the variability of the analysis, a standardized position of the animal in the microCT scanning bed and defined acquisition settings will be highly recommendable, as suggested in a previous publication for semi-automatic segmentations [16]. Birk et al. designed an automatized method for lung tissue segmentation based on histogram levels. For the machine learning process, all the animals were scanned in a standardized position for an easy recognition of the different structures. Following this reasoning, we defined a standard positioning of the animals during the microCT acquisitions and the image analysis as well as a calibration of the equipment before the scans. In the case of threshold segmentation, these protocols and calibrations allow for working with an accurate window level for thresholding, such as in clinical imaging. In fact, the intensity window applied in this project (−700/−300 HU) is the same as in previous studies in human medicine [8,12,19] or preclinical imaging [33] although some other authors used different HU values for lung and air thresholds, such as Elgeti et al., who used a wider range (−950/−280 HU). Ruscitti and collaborators included as lung tissue all the voxels within −900 and −500 HU. They maintain the idea that the rodent lung tissue has different radiodensity and HU values than the human one and defined the mean value of their mice studies as −575 HU instead of the human −690 HU [34,35]. Wang, Li and Li tried different values and finally selected the (−700/−300 HU) range [8]. In a similar way, we checked two potential window levels and decided to use the (−700/−300 HU) range.

Regarding the technical part of the experiment, the acquisition protocol used during this experiment is similar to that described in other rodent imaging publications [11,14,16,24,32,33,34,35,36,37]. Regarding the results for the negative control animals in the different pathologic models, no differences were observed after comparing the results from the different used microCT systems. This fact lets us to assume the potential applicability of the presented protocol for studies acquired in different imaging systems, even retrospectively. The whole protocol is based on anatomical references and thoracic morphology, but is not significantly affected by technical parameters as spatial resolution or X-ray energies. In theory, the protocol could be applied in every study that accomplishes the minimum requirements to assess the anatomical recognition of the defined landmarks and segmentations. Regarding the quality of the images, the studies with movement artifacts should be repeated for accurate results. In case of a retrospective use of the protocol, studies with movement artifacts that affect the pixel intensity should be discarded for obtaining the mathematical correlation between thoracic and lung delineated volumes.

The sensibility and specificity results were 74% for sensibility and 78% for specificity.

Based on previous publications, pathologic animals with severe disease will increase their lung volume as a response to functional lung tissue reduction during pathology progression [14,32]. This will affect all the volumes measured during this project, including thoracic volume, manually delineated volume, thresholded volume and theoretical lung volume. This last will be affected because it is calculated indirectly from thoracic and manually delineated volumes, and both of them should change with the development of different lung pathologies. Vande Velde and collaborators found a rise of 24% in total lung volume of the volume after 7 days of bleomycin instillation and 28–32% at the 28th day. The thoracic volume was not measured by Vande Velde. In our experiment, we compared the manual lung volumes between negative control and pathologic animals in a different way from Vande Velde, who made a longitudinal study before and after inducing the lung pathology. In relation to threshold aerated lung volume, the values also changed, but in this case with a reduction in volume due to the pathology, which affects the lung tissue density. In another publication, Barck et al. found an increase in chest volume during the development of lung tumors, but no changes were observed in air space volume [14]. Comparing to their results, the transgenic lung tumor model presents a similar tendency, with less anatomical and adaptive changes of different volumes than the more aggressive pathologies where a major percentage of lung tissue is affected. The lowest value in aerated lung volume (functional lung tissue) corresponds to fungal infection, followed by the metastatic model and transgenic tumor model. In opposition, manually delineated lung volume as well as thoracic volume are higher in the fungal model, followed by the metastatic model and transgenic tumor model. The results obtained in our project follow the previous published changes in the different lung volumes. These results give support to the previously published findings of adaptative responses to lung pathologies [2,14].

After confirming that our results follow the trends of the previous publications regarding the lung volume changes in pathologic models, we analyze the possible effect of the pathology in our theoretical lung volume calculation. For that purpose, we compared the deviation of the theorical value from the gold standard, in this case the manually delineated lung volume because of the effect of the pathology on the thresholded volume. The mean deviation of the values between theoretical lung volume and manual delineation in pathologic animals reaches 7.46%, while the negative control animals from the same animal models have a mean deviation of 6.43%, bringing to light the independence of theoretical lung volume from the presence or absence of lung pathology. This is another point that supports the applicability of the presented protocol in lung pathology research.

## 5. Conclusions

The present manuscript proposed a fast and accurate method for lung volume calculations. Compared to the gold standard methodology (manual delineation of the lungs), this protocol can save 60 to 75% of the time consumed in lung volume calculation and its reproducibility and replicability is higher, reducing the operator dependency of the results. These two points (reduced time of analysis and operator dependency) are key points when using imaging technologies in lung research. The use of this protocol can improve the quality of the imaging experiments and reduce the time spent in this part of the projects.

This volume calculation is based on thorax anatomy and morphology, so lung pathologies that modify tissue radiodensity will not affect this protocol but thoracic anatomical modifications could do so. Correct positioning of animals during scanning and posterior reorientation of the studies are essential.

This protocol is applicable in any animal model and strain, but it should be recommendable to obtain specific correlation formulas for each model due to the potential for specific animal model variability.

Machine learning and automatized processes could improve the efficacy and usefulness of the presented protocol. Automatic recognition of anatomical landmarks and subsequent immediate calculation of thoracic volume would mean a solution for time consumption during imaging analysis in lung research. Complementary studies are necessary for the design and development of automatized protocols.

## Figures and Tables

**Figure 1 jimaging-08-00204-f001:**
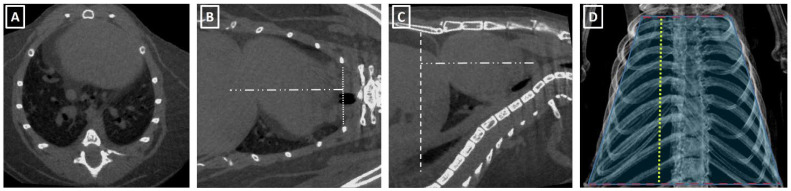
Thoracic volume measuring process. (**A**) Axial reorientation of the thorax. (**B**) Short basis measure line placement at carina level (dot line) and height distance measurement (stripes and dots line) in coronal view. (**C**) Height measure (stripes and dots line) and big basis line placement at diaphragmatic level (stripes line). (**D**) Representative figure of thoracic truncated cone overlaid on thorax µCT.

**Figure 2 jimaging-08-00204-f002:**
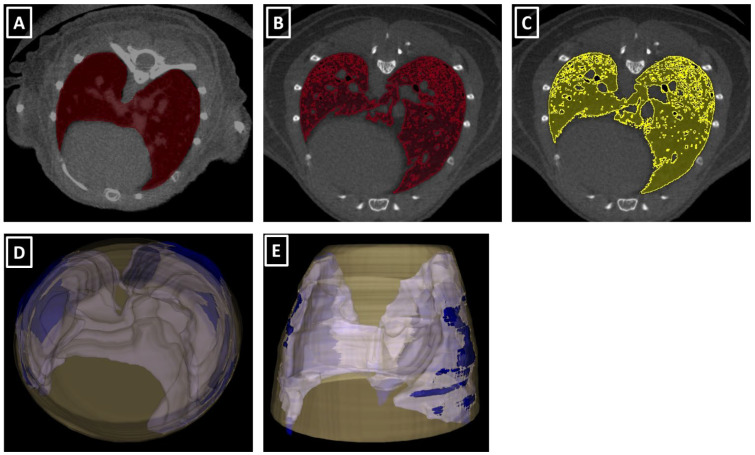
Manually delineated and thresholded lung volume segmentations. (**A**) Manual delineation of lung volume, with ROI (red) covering the lung overlaid on transversal µCT image of the lung. Note the inclusion of intrapulmonary soft tissue. (**B**) Thresholded segmentation of aerated lung tissue (red ROI) applying a −700/−400 Hounsfield Units threshold. (**C**) Thresholded segmentation of aerated lung tissue (yellow ROI) with −700/−300 Hounsfield Units threshold. Note the different lung tissue inclusion in the segmentation between (**B**) and (**C**), depending on the applied threshold. (**D**,**E**) Three-dimensional rendering of the manually delineated segmentation (blue) and the thoracic volume superimposed (yellow). The inclusion of structures such as the heart in the thoracic volume is visually evident.

**Figure 3 jimaging-08-00204-f003:**
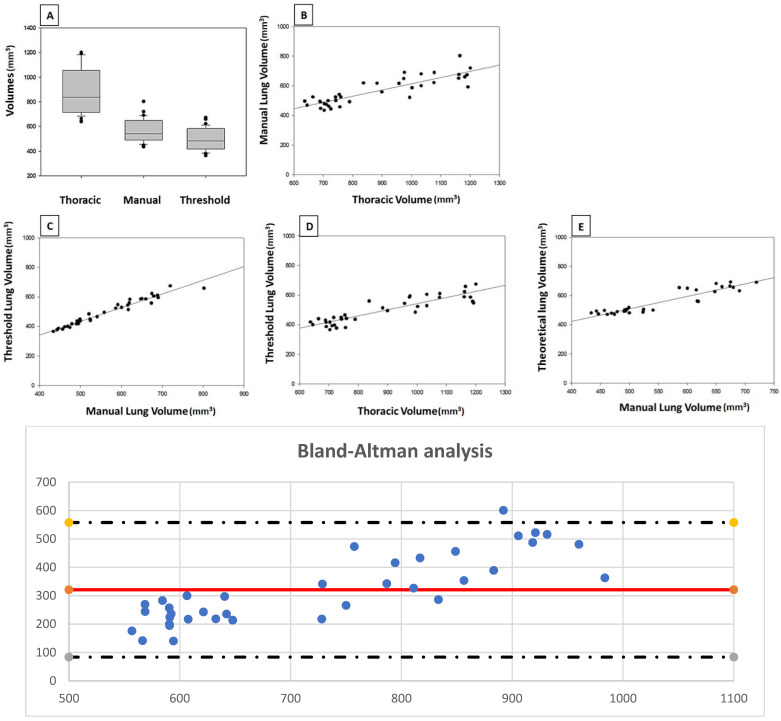
Thoracic, manually delineated and thresholded volumes from healthy mice group and Bland-Altman plot. (**A**) Distribution of mean values. (**B**) Correlation between thoracic and manually delineated lung volumes (R^2^ = 0.7126). (**C**) Correlation between manual and threshold volumes (R^2^ = 0.9526). (**D**) Correlation between thoracic and threshold lung volumes (R^2^ = 0.6871). (**E**) Correlation between manually delineated lung volume and theoretical lung volume (R^2^ = 0.8573). In all the cases, the correlation is statistically significant (*p*-values shown in Table A3). At the bottom, the Bland-Altman plot, where all the points except one are included in the range of 95% of confidence for the theoretical volume minus gold standard volume. The *x*-axis shows the average measurement of lung volume using both methods while *y*-axis shows the difference between the theoretical lung volume and manually delineated lung volume. The red line represents the average difference in measurements.

**Figure 4 jimaging-08-00204-f004:**
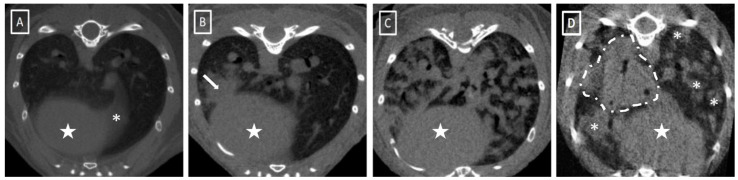
Representative lung microCT images of the different animal models used. (**A**) Healthy animal. Note the absence of intrapulmonary soft tissue structures except vessels, heart (white star) and the first part of the diaphragm (white asterisk). (**B**) Transgenic lung tumor model with a solitary lung tumor (white arrow) in contact with heart burden (white star). (**C**) Metastatic lung cancer model. Presence of multiple soft tissue structures compatible with lung tumors. (**D**) Fungal infection model. Presence of multiple hyperdense structures (fungal abscesses, asterisks) and a consolidated lung tissue dorsal to the heart (area delimited by a white dashed line).

**Figure 5 jimaging-08-00204-f005:**
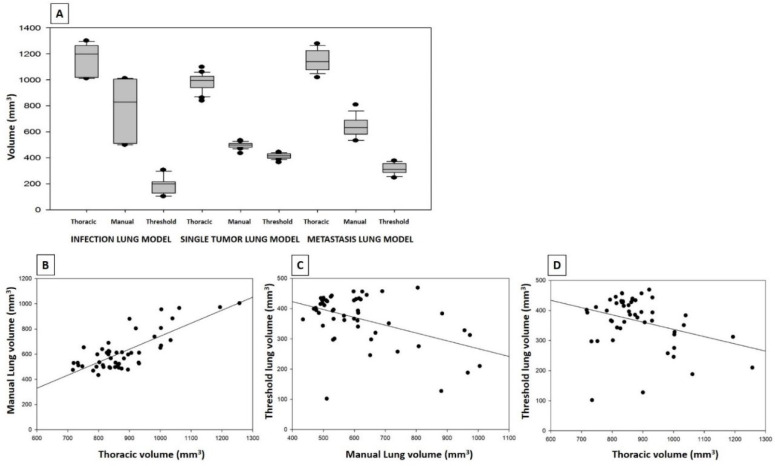
Thoracic, manually delineated and thresholded lung volumes of pathologic groups. (**A**) Mean values of volumes from the three calculations in metastatic model, transgenic lung tumor model and fungal infection model) (**B**) Correlation between thoracic and manually delineated lung volumes including the data from the different animal models (mm^3^), R^2^: 0.619. (**C**) Correlation between manually delineated and thresholded lung volumes in the same batch of data (mm^3^), R^2^: 0.242. (**D**) Correlation between thoracic and threshold lung volumes (mm^3^), R^2^: 0.038. Note the lack of correlation between threshold volume and the other volumes, while there is a significant correlation between thoracic and manually delineated volumes, as in the healthy group. All the statistical values are displayed in Table A3.

**Table 1 jimaging-08-00204-t001:** Results from pathologic animals in the different groups. Values are in mm^3^ and Standard Deviation in brackets.

		Thoracic Volume	Manual Volume	Threshold Volume	Theoretical Volume
Metastatic lung cancer model	Pathologic animals	893.20 (109.26)	640.58 (83.06)	315.92 (43.67)	635.28 (57.81)
	Negative control	841.88 (53.58)	602.27 (30.22)	419.7 (31.24)	608.13 (28.35)
Transgenic lung cancer model	Pathologic animals	860.12 (61.69)	503.91 (19.31)	408.72 (25.04)	524.30 (36.05)
	Negative control	829.95 (57.70)	523.11 (102.99)	418.43 (29.28)	506.75 (33.57)
Fungal infection model	Pathologic animals	1027.29 (175.36)	882.42 (170.26)	236.00 (107.17)	863.102 (168.49)
	Negative control	823.99 (11.29)	594.166 (135.76)	400.44 (80.71)	661.924 (134.35)
		Ratio Manual/Thoracic	Ratio Threshold/ Manual	Ratio Threshold/ Thoracic	
Metastatic lung cancer model	Pathologic animals	71.97 (6.90)	50.34 (10.63)	35.98 (7.03)	
	Negative control	71.69 (3.95)	69.72 (4.37)	50.02 (4.59)	
Transgenic lung cancer model	Pathologic animals	58.83 (4.46)	81.20 (5.40)	47.79 (5.10)	
	Negative control	62.82 (9.26)	81.45 (8.94)	50.51 (3.24)	
Fungal infection model	Pathologic animals	85.74 (9.82)	26.38 (10.39)	22.61 (9.36)	
	Negative control	72 (15.49)	67.61 (1.86)	48.53 (9.13)	

## Data Availability

Not applicable.

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
