# Peer review of "Lung Volume Calculation in Preclinical MicroCT: A Fast Geometrical Approach"

_2313-433X, 2022, doi:10.3390/jimaging8080204_

Round 1
Reviewer 1 Report
Here, Juan Antonio Camara et al. developed an alternative mathematical approach, referred to threshold volume in the manuscript, that can accurately measure lung volume by micro-CT. Authors first show how much faster this approach is compared to previous methodologies without compromising reproducibility and repeatability. After evaluating the lung volume from healthy mice, authors used three different disease models to evaluate the capabilities of the mathematical approach.
The manuscript represents a rigorous and straightforward scientific work. The approach presented here can facilitate the data acquisition of large animal studies as well as enabling the comparison of studies between different institutions around the world. I recommend the publication of this manuscript in its present form.
Author Response
Thank you very much for your comments. We did our best in this project.
Best regards,
Reviewer 2 Report
1) In methodology section, author acknowledges free breathing during acquisition. It would be important to put all images in comparable mode of either complete inspiration or expiration. This is especially important as the paper is about correlation of thoracic volume and lung volume.
2) Author has mentioned primary lung tumor, metastasis and fundal infection as the test models. However, these models include space occupying lesions with substraction from aerated lung volume. Author should at least give one example of COPD / emphysema where aerated lung volume changes.
3) Author should include 3D lung and thoracic volume images along with 2D X-ray and CT images, to clarify the differences.
4) If the author can provide a sensitivity and specificity analysis of this methodology, scientific implications will improve.
Author Response
Thank you very much for your comments. We really appreciate your suggestions.
Please see the attachment for the detailed responses.
Best regards,

Reviewer 3 Report
The authors present a method for lung volume calculation in preclinical animal models, including models of lung disease. The aim of the work was to generate a method which robustly calculates lung volume (as a measure of health/disease) with reduced observer-bias and analysis time. The authors have identified a challenge in preclinical imaging which would benefit from robust analytical methodology, in particular since there are often not trained radiologists specific to the preclinical imaging community. The simplicity of this technique is a great strength, and would be easy to implement in nearly any setting. Robustness to inter-observer variability is also a strength. However, there are some concerns which limit enthusiasm for the current version. The major concerns are primarily about how the animals were reported, and the statistical method employed. Correcting these concerns should generally be achievable, and should improve the accessibility and interpretation of the results.
Major Concerns:
1) There is much confusion surrounding the animals/images included in the analyses. The methods state numbers of mice (37 healthy, 23 lung met model, 18 primary lung tumor model, and 9 lung infection model). However, the methods also state that "for each pathologic model, sham-induced age- and gender matched control animals were included". How many of these animals were included? Please clarify how many animals of each condition (including sham) were included on study.
2) The number of animals listed does not appear to match the number of data points in the results. There appear to be fewer animals reported in the results. Please explain and justify this discrepancy.
3) In the methods, the authors state "Upon prior quality control, scans with artifacts were discarded." Is this the reason for the discrepancy in animal numbers and data points? If so, there needs to be significantly more information about exclusion, and these exclusions must be explicitly justified. If the authors are referring to a specific protocol for quality control, it should be cited/described. I would recommend explicit descriptions of why data was excluded, and state the number of images excluded for each reason. If exclusion of certain types of artifacts is required for the success of the proposed algorithm, this needs to be explicitly discussed as a conditional limitation.
4) The use of linear regression to compare methods is not typically how we compare the success of a novel imaging analysis technique with gold standards. It would be informative to perform a Bland-Altman analysis and report these plots. Bland-Altman will allow the authors to state that the new method is comparable to the gold standard while informing of any size-related bias (which the authors touched on but did not provide further results). This is the appropriate way to show the utility of a new method.
5) What do the authors mean by "negative animals". This is not defined in the methods, and thus I don't know how to interpret all of the results reported in "negative animals".
6) It appears that the thoracic model does not outperform the thresholded method in the primary lung tumor model. The most consistency between the thoracic method and the gold standard was in the lung infection model, which was performed on a different scanner from all other animals. This discrepancy is not addressed in the manuscript.
7) In the discussion (line 364) there is a lengthy discussion of changes in lung volume due to pathology. This was not reported as results, and was generally not included as an aim of the study. Should this be included as a result/figure in the main body?
8) Similar to the previous comment, there is a discussion of variance at the very end of the discussion. These data should be reported in the results, and methods for comparing variance need to be explicitly stated in the methods.
Minor Concerns:
9) The results reported on page 8 (first paragraph, line 238) are extremely difficult to interpret as written. I would recommend a table instead of listing them.
10) The results would be easier to interpret if the gold standard "manual" was always reported first or last. It may even help to have the gold standard appear as a different color/shade, so readers know what ground truth is.
11) The authors should include in the methods if images were respiratory gated.
12) In the discussion, the authors state that both the manual segmentation and thresholding were used as gold standards. Are thresholded images truly accepted as a "gold standard"? The paper also suggests that thresholding is not a particularly good method in pathologic tissues. Does comparison with Bland-Altman of the new methods vs. manual, and thresholding vs. manual suggest that the new method is superior to thresholding in pathologic tissues?
13) Lines 307-320 (plus citations) would be more appropriate for the intro.
14) It would be helpful if the authors could include reproducibility information about the image acquisition. For example, if you scan the same sample twice, how reproducible are the results? Not required, but would strengthen the argument.
15) In the discussion, the authors mention that there was no difference between scanners, but this data was not specifically described in the results, so interpretation is unclear.
16) Are there plans to validate the findings in a different healthy group?
Overall Impressions:
I think there are some very interesting elements presented in this paper that could be valuable to the preclinical imaging community as a whole. I clearly understand the stated problem, and appreciate the simplicity of the approach. The most prominent issue that needs to be addressed is the issue of how the animals are reported in the results, and how data were excluded. I would then strongly encourage the use of a Bland-Altman analysis over a linear regression, as this is how we typically justify that new methods are as good as a gold standard. If the concerns are adequately addressed, I feel this paper will be an interesting addition to preclinical imaging methodology.
Author Response

(The authors gave the same response as above.)

Round 2
Reviewer 3 Report
The revised manuscript has undergone targeted changes that have addressed all of my original comments. The expanded detail on animal distribution and inclusion of Bland-Altman are exceptionally helpful, and make the impact of this work much easier to interpret. I appreciate the efforts made by the authors, and think that the work will be very useful to preclinical imagers performing CT.
I have two small notes that are strictly formatting. Figure 3 appears to have repeated panels by mistake. It would also be appreciated if the authors could include axis labels on the Bland-Altman plot in Figure 3. Again, these are very minor changes.
Excellent work.